# On the Way to Accounting for Lung Modulation Effects in Particle Therapy of Lung Cancer Patients—A Review

**DOI:** 10.3390/cancers16213598

**Published:** 2024-10-25

**Authors:** Matthias Witt, Uli Weber, Veronika Flatten, Jessica Stolzenberg, Rita Engenhart-Cabillic, Klemens Zink, Kilian-Simon Baumann

**Affiliations:** 1Institute of Medical Physics and Radiation Protection, University of Applied Sciences, 35390 Giessen, Germany; u.weber@gsi.de (U.W.); jessica.stolzenberg@lse.thm.de (J.S.); klemens.zink@lse.thm.de (K.Z.); kilian-simon.baumann@lse.thm.de (K.-S.B.); 2Department of Radiotherapy and Radiation Oncology, Marburg University Hospital, 35043 Marburg, Germany; rita.engenhart-cabillic@uk-gm.de; 3Marburg Ion-Beam Therapy Center (MIT), 35043 Marburg, Germany; 4Biophysics Division, GSI Helmholtzzentrum fuer Schwerionenforschung, 64291 Darmstadt, Germany; 5LOEWE Research Cluster for Advanced Medical Physics in Imaging and Therapy (ADMIT), TH Mittelhessen University of Applied Sciences, 35390 Giessen, Germany; 6Sun Nuclear, Mirion Medical Company, Melbourne, FL 32940, USA; flatten@mirion.com

**Keywords:** particle therapy, lung cancer, lung modulation effects, dose degradation, dose calculation, Monte Carlo, treatment planning

## Abstract

This research aims to improve the precision of particle therapy for lung cancer treatment. The heterogeneous, microscopic structure of lung tissue leads to a broadening of the very sharp dose profiles of protons and other light ions. This can result in higher doses in healthy tissue as well as a reduced dose in the target volume and thereby reducing the potential benefit of particle therapy for lung cancer. This review summarizes existing models that account for these lung tissue effects and explore how they can be better integrated into treatment planning. By taking into account these so called lung modulation effects, it is possible to target tumors more precisely while minimizing harm to surrounding healthy tissues, ultimately benefiting cancer patients.

## 1. Introduction

Particle therapy is a promising alternative to conventional photon therapy when treating non-small cell lung cancer (NSCLC) [1,2]. In particular, the sharp distal dose fall-off allows for a better sparing of healthy tissue while delivering a high dose to the target. However, the heterogeneous structure of lung tissue modulates the dose distribution, especially a degradation of the distal dose fall-off. This can lead to an underdosage of the target volume and an overdosage of distal normal tissue, if the effect is not considered during treatment planning [3,4,5,6]. Historically, accounting for the microscopic structure of lung tissue in treatment planning was not feasible, as typical planning CT systems lacked the necessary resolution. Only in recent years have alternative approaches been developed to address these modulating effects. This review provides a comprehensive overview of the models developed to account for these modulation effects. It summarizes studies focused on determining modulation power as a predictor of lung modulation, based on ex vivo porcine lung measurements [7,8], lung surrogate materials [9], and the first study to determine modulation power using clinical CT images [10]. A detailed comparison of the two primary models developed to account for lung modulation is provided [11,12,13,14], along with a discussion of the advantages and limitations of each. The review covers early investigations into dose uncertainties due to lung modulation effects in CT-based lung phantoms [15], as well as clinical treatment plans for proton and carbon ion irradiation [12,13,14,16]. Additionally, the review discusses future challenges in integrating these solutions into routine clinical treatment planning.

## 2. Particle Therapy for Lung Cancer Patients—Benefits and Challenges

Since the radiological use of fast protons was proposed in 1946 by Wilson [17], the significance of protons, helium ions, and other light ions, such as carbon ions, as a treatment modality for cancer patients has been continuously growing worldwide. Due to their physical properties, protons and light ions can be used to deliver a high dose to the target while reducing the dose deposited in surrounding normal tissue compared to conventional photon therapy. To deliver protons and light ions, two techniques are available: passive scattering and active scanning. Passive scattering utilizes passive beam elements, such as scattering foils, collimators, compensators, and modulator wheels, to shape the beam according to the patient-specific anatomy. In contrast, active scanning delivers the dose to the target using pencil beams, scanning the tumor in depth (by varying particle energy) and laterally (via deflection with dipole magnets). While active scanning is more time-consuming than passive scattering, the production of secondary particles, particularly neutrons, is up to ten times lower since almost no passive beam elements are used [18,19]. Additionally, the total dose to the body (integral dose) is lower with active scanning, and intensity-modulated particle therapy is possible [20,21]. The lower out-of-field dose and the possibility of achieving higher dose conformity [22] can be exploited for the treatment of NSCLC [1,2]. In general, NSCLC demonstrates poor local control, and depending on the stage of the tumor, the target volume can be in proximity to organs at risk, such as the esophagus, trachea, heart, and spinal cord, potentially restricting the effectiveness of conventional therapy [2]. With particle therapy, the dose in the target volume can be escalated significantly [1,23], and the integral dose can be reduced even compared to modern IMRT [24,25,26,27,28,29]. This is especially true for locally advanced (stage III) NSCLC, which typically involves large primary tumors located near critical structures; particle therapy can increase median survival and reduce toxicities [1,2].

However, although promising, particle therapy for NSCLC is associated with some of the most challenging aspects of state-of-the-art treatment routines. The main challenge arises from the finite range of particles, which strongly depends on the traversed medium. As a result, changes in the patient’s anatomy can lead to significant dose uncertainties if not properly accounted for. Intrafractional motion plays an important role when treating lung tumors due to breathing movement. On one hand, breathing can lead to changes in the position of the ribs relative to the beam, drastically influencing the range of the particles and, consequently, dose deposition. On the other hand, when using the active scanning technique, breathing can create interference between the motion of the target and the movement of the pencil beam that scans the target volume. This interference leads to the so-called interplay effect [30], resulting in reduced dose coverage of the target and the creation of hot and cold spots. Therefore, motion mitigation techniques are necessary, such as robust planning, gating, rescanning, or tracking [31,32,33,34,35].

## 3. Modulation Effects of Lung Tissue

Another challenge in particle therapy for lung cancer patients arises from the structure of the lung itself. The lung consists of tubular branches—the bronchi—that are necessary for transporting inhaled air into the lung. On their way into the lung, these bronchi subdivide into smaller structures, the so-called bronchioles. These bronchioles then end in microscopic clusters of air-filled sacs, the alveoli. The alveoli consist of an epithelial layer and an extracellular matrix surrounded by capillaries. Through this respiratory membrane, gas exchange is facilitated. The size of the alveoli ranges from ∼50 μm to ∼250 μm between exhalation and inhalation [36,37]. While the alveoli are filled with air, which has a physical density of ∼1 mg/cm^3^, the physical density of the surrounding tissue is similar to water, at around 1 g/cm^3^. Hence, on a microscopic scale, the lung exhibits density heterogeneity. This microscopic density heterogeneity influences the range of the particles traversing the lung tissue. Depending on their path through the lung, the particles experience different compositions of low-density alveoli and high-density tissue regions. These varying density compositions lead to differences in energy losses and, consequently, the ranges of the particles. In conventional photon or electron radiotherapy, the maximum of the depth–dose distribution is much broader compared to the peak of proton or carbon beams. The same is true for other high-LET modalities, such as boron neutron capture therapy (BNCT) and gadolinium neutron capture therapy (GdBNCT) [38,39,40]. Since the half-width of their broad maximum is many times larger than the modulation range in porous lung tissue, there is no relevant change in the depth–dose profile of the primary neutrons due to porosity. Furthermore, the secondary alpha particles in BNCT have a very short range of about 5 μm; therefore, the modulation does not affect the ranges [38]. This changes dramatically for particle beams such as proton and carbon beams, where the initial sharp Bragg peak is significantly broadened (as shown in this paper), and consequently, the distal dose fall-off is degraded. This modulation of the Bragg peak is also referred to as degradation of the Bragg peak or lung modulation. An example is shown in Figure 1 for 114 MeV protons in water. The black solid line represents the reference Bragg peak, while the black dashed line depicts the same Bragg peak after traversing an additional 20 mm of water. The modulated depth–dose distributions, shown in blue and red, represent the results after traversing 20 mm of two different heterogeneous materials. The modulating material leads to a shift of the Bragg peak due to the additional material in the beam path and a broadening of the Bragg peak due to the heterogeneous structure. This modulating property of heterogeneous materials was already described in 1986 by Urie [41]: by measuring dose distributions in a water phantom, it was demonstrated that the full width at half-maximum (FWHM) of the Bragg peak, as well as the distal dose fall-off, increased when heterogeneous materials were positioned in the particle beam. Sawakuchi et al. [4] investigated the Bragg peak degradation using Monte Carlo simulations with virtual geometries consisting of bone and air voxels. They examined the change in the energy spectrum of protons due to different energy losses and proposed a model to predict the change in the distal dose fall-off depending on the energy spectrum.

In the clinical context, the Bragg peak degradation due to heterogeneous lung tissue can significantly influence the dose distribution in lung cancer patients [3]. Specifically, the degradation can lead to an underdosage of the target volume and an overdosage of distal normal tissue if particles are traversing lung tissue on their way to the target [4,5,6]. Accordingly, the effects of Bragg peak degradation should be considered during the treatment planning process [6]. However, typical CT scanners in the clinic do not have sufficient resolution to fully resolve the microscopic structure of the lung tissue. Due to the restricted resolution, microscopic structures are merged into larger voxels [3,6]. As a result, a more homogeneous density distribution within the lung is predicted, making it impossible to directly consider Bragg peak degradation effects during treatment planning on the basis of clinical CT images. It was shown that averaging over densities in CT images can lead to differences in the Bragg peak dose of up to 11% and uncertainties in the distal edge degradation of up to 1.1 mm for clinical proton beams [42].

These effects were further investigated by Titt et al. [6]. A high-resolution 3D-printed phantom was used as a lung substitute. The phantom was built from microscopic voxels consisting of either plastic or air. Depth dose distributions of 150 MeV and 200 MeV protons in a water phantom downstream of the phantom were measured as well as simulated with the Monte Carlo method. It was shown that the heterogeneous structure of the phantom led to a degradation of the Bragg peak. Additionally, CT images of the 3D-printed phantom were acquired, and subsequently, depth–dose distributions of protons penetrating the CT images were simulated and compared to the simulations of the original 3D-printed phantom. In the simulations based on the CT images of the phantom, the distal dose fall-off was found to be underestimated by a maximum of 2 mm in water (equivalent to approximately 10 mm in lung tissue). Additionally, the dose in the Bragg peak was observed to be overestimated by a maximum of 35% [6]. Furthermore, Titt et al. investigated the Bragg peak degradation using a plastinated human lung. Irradiation through the lung resulted in a distal fall-off width of the Bragg curve in water that was increased by up to 60% compared to an unperturbed reference curve. Furthermore, the authors put forth a mathematical model to describe the effects of Bragg peak degradation. The degraded dose distribution could be estimated by shifting an unperturbed pristine dose distribution by the water-equivalent thickness of the phantom and applying a convolution with a normal distribution.

## 4. The Concept of Modulation Power

The concept of describing the degraded depth–dose distribution downstream from heterogeneous materials by shifting an unperturbed reference curve and convolving it with a normal distribution was further optimized by Ringbæk et al. [9]. By including the shift due to the water-equivalent thickness of the heterogeneous material directly into the normal distribution (μ≠0), the degraded dose distribution could be estimated immediately by convolving the unperturbed pristine dose distribution with that adapted normal distribution (without the need to additionally shift the unperturbed dose distribution). The authors further developed a mathematical model to derive this normal distribution starting from a voxelized geometry (see Figure 2). The voxels consisted of either a high-density material with density $\rho_{\mathrm{med}}$ or a low-density material with density 0 g/cm^3^ (in a first approximation, the air in lung tissue can be replaced by vacuum in terms of the modulation effects). The voxels were placed randomly within the geometry, while the probability that a voxel consisted of the high-density material was set to *p*. Hence, an average density $\rho_{\mathrm{mean}}$ of this geometry could be derived:(1)$$\rho_{\mathrm{mean}}=p\cdot \rho_{\mathrm{med}}$$

A particle traversing this geometry would hit a specific number *k* of voxels consisting of the high-density material depending on the path the particle took through the geometry. The corresponding probability of hitting *k* voxels could be calculated using a binomial distribution that devolved into a normal distribution if the geometry was large enough (that is typically the case in the context of lung irradiation). The number *k* of voxels could further be substituted by a water-equivalent thickness using the density $\rho_{\mathrm{med}}$ and the length *d* of the voxel in beam direction. Hence, the probability that a certain path through the voxelized geometry had a water-equivalent thickness $t^{\prime }$ could be calculated as follows:(2)$$P\left(t^{\prime }|\sigma ,t\right)=\frac{1}{\sqrt{2\pi \sigma^{2}}}\exp\left(\frac{{\left(t^{\prime }-t\right)}^{2}}{2\sigma^{2}}\right)$$
where *t* is the average water-equivalent thickness of the voxelized geometry, and σ describes the broadening strength of the modulating material. From the parameters of the voxelized geometry, σ could be calculated directly as follows:(3)$$\sigma =t\cdot d\cdot \rho_{\mathrm{med}}\left(1-p\right)$$

Since dose distributions are mostly measured in water, it was reasonable to define σ in a water-equivalent unit. Hence, *t* as well as $\rho_{\mathrm{med}}$ had to be interpreted as water-equivalent quantities.

The normal distribution $P\left(t^{\prime }|\sigma ,t\right)$ from Equation (Equation 2) was then used to derive the modulated dose distribution $b_{*}\left(z\right)$ in water downstream from this voxelized geometry by convolving $P\left(t^{\prime }|\sigma ,t\right)$ with an unperturbed reference curve $b_{0}\left(z\right)$, while *z* is the depth in water:(4)$$b_{*}\left(z\right)=\left(P\left(t^{\prime }|\sigma ,t\right)\;*\;b_{0}\right)\left(z\right)=\int_{-\infty }^{\infty }P\left(t^{\prime }|\sigma ,t\right)b_{0}\left(z+t^{\prime }\right)dt^{\prime }$$

A visualization of this concept is shown in Figure 2. It should be noted that the shift of the unperturbed reference curve does not refer to the depth of the Bragg peak itself, but rather to the depth $z_{p82}$ where the dose of the distal fall-off is ∼82% of the maximum dose, as this corresponds to the mean range of particles [43].

Using this mathematical description of the Bragg peak degradation, Ringbæk et al. [9] introduced a new material characteristic for heterogeneous materials, the modulation power $P_{\mathrm{mod}}$. It can be calculated from *t* and σ as follows:(5)$$P_{\mathrm{mod}}=\frac{\sigma^{2}}{t}=d\left(\rho_{\mathrm{med}}-\rho_{\mathrm{mean}}\right)$$

$P_{\mathrm{mod}}$ is a quantification of the modulation strength of a heterogeneous material and is given in the units of water-equivalent length or mass per area. The modulation power is independent on the geometrical thickness of the heterogeneous material and solely depends on the structure size *d* of the heterogeneous material and the density differences. If the exact structure and density distribution of a heterogeneous material is known, $P_{\mathrm{mod}}$ can be calculated directly. If no knowledge about the structure size and density distribution is available, as it is the case for lung tissue, $P_{\mathrm{mod}}$ can be measured with in-beam experiments: both an unperturbed reference curve $b_{0}\left(z\right)$ and the degraded depth–dose distribution $b_{*}\left(z\right)$ have to be measured, e.g., in a water column. Subsequently, *t* and σ of the normal distribution $P\left(t^{\prime }|\sigma ,t\right)$ have to be optimized to minimize the difference between $b_{*}\left(z\right)$ and the convolution $\left(P\left(t^{\prime }|\sigma ,t\right)\;*\;b_{0}\right)\left(z\right)$. $P_{\mathrm{mod}}$ can then be derived from *t* and σ.

## 5. Experimental Determination of Modulation Power

Ringbæk et al. [9] used this method to measure the modulation power of various heterogeneous, inorganic materials like Polystyrol foam, Gammex LN300, Gammex LN400, and several ripple filters [44,45] as well as an ex vivo porcine lung sample. Modulation power values for inorganic materials ranged from 5 μm for very fine Polystyrol to 350 μm for Gammex LN300. The modulation power for the ex vivo porcine lung sample was 530 μm. Hranek et al. [46] also determined $P_{\mathrm{mod}}$ for inorganic heterogeneous lung substitutes including cork, konjac sponge, floral foam, and commercial lung tissue-equivalent plates (LTEP). $P_{\mathrm{mod}}$ was in the range of 100 μm to over 1000 μm. To further determine $P_{\mathrm{mod}}$ for ex vivo porcine lung tissue, Witt [8] and Burg et al. [7] investigated ex vivo lung tissue in carbon ion beams. Witt [8] determined the modulation power of complete porcine lungs to be in the range from 300 μm to 750 μm. The lungs were kept in different ventilation states and were irradiated at different positions. Since complete lungs were considered, the determined $P_{\mathrm{mod}}$ corresponded to an integrated modulation power of all different structures arranged in the beam line. Since $P_{\mathrm{mod}}$ depends on the size of the structure, it varies with the position in the lung. In the peripheral region of the lung, the structures are generally smaller resulting in smaller modulation power values, while the structures at the center of the lung are larger resulting in larger modulation power values. For almost all measurements, $P_{\mathrm{mod}}$ was in the range between 300 μm to 500 μm while the average modulation power was 450 μm. For one measurement, $P_{\mathrm{mod}}$ was 750 μm. However, in that measurement, the particle beam traversed the lung at a position where a large bronchial structure was located.

Burg et al. [7] determined $P_{\mathrm{mod}}$ for smaller lung samples (only 4 cm in beam direction). Still, an integrated $P_{\mathrm{mod}}$ was determined, on a much smaller scale, making the results more precise. The lung samples originated from domestic pigs as well as wild boars frozen in a ventilated state. Subsequently, samples of 4×4×10 cm^3^ in size were cut from the frozen lungs and investigated in carbon ion beams (see Figure 3).

Results for $P_{\mathrm{mod}}$ were in the range between 60 μm and 370 μm for the lung tissue from domestic pigs. For wild boar’s lung tissue, $P_{\mathrm{mod}}$ was larger with up to 580 μm. The authors discussed that higher values of $P_{\mathrm{mod}}$ for wild boar’s lung tissue were due to a higher blood content within the lung that led to a higher physical density. It was shown that for some lung samples, a fit with a normal distribution was not able to reproduce the broadened depth–dose distribution. Especially for samples where larger structures were located in the beam, a fit with two normal distributions was necessary to reproduce the broadened depth–dose distribution. The authors discussed that larger structures also led to modulating effects that overlaid the modulating effect of the residual lung tissue. To further investigate this connection, lung samples were imaged in a micro-CT with a resolution of 50 μm. It could be shown that for samples that needed two normal distributions, larger structures could be seen in the micro-CT image.

To investigate whether the modulation power values measured for porcine lung tissue were applicable to human lung tissue, Witt [8] and Baumann et al. [11] investigated high-resolution CT images of human lung tissue samples (see Figure 4). Lung samples were prepared using a “critical point drying” method [47] and CT images were acquired with a resolution of 4 μm. The values of $P_{\mathrm{mod}}$ were in the range from 50 μm to 250 μm and hence smaller compared to in-beam measurements of porcine lung tissue. However, the “critical point drying” method used for preparation led to a loss of water of up to 37% that significantly reduced the size of structures in lung tissue affecting the determined modulation power.

In conclusion, the measurement of $P_{\mathrm{mod}}$ for lung tissue is still a challenging task. In particular, the preparation of lung tissue samples influences the properties of the tissue structure. Either by a loss of blood in the case of ex vivo porcine lungs or by a loss of water for the preparation of human ex vivo lung tissue. Additionally, preparation of lung tissue samples can cause mechanical damage to the tissue. Since $P_{\mathrm{mod}}$ strongly depends on the structure’s size, measurements of ex vivo lung tissue are connected to correspondingly large uncertainties. Hence, measurements with ex vivo lung tissue samples can only be used to estimate the order of the modulation power of human lung tissue. Additionally, the data situation is not sufficient to clearly quantify the dependence of $P_{\mathrm{mod}}$ on the position within the lung and the respiratory state. Both parameters are assumed to influence the modulation power since they affect the structure size and density distribution within the lung. Another important point to consider is that to date, only lung samples from healthy subjects have been examined. It is highly likely that pathological conditions affecting the lungs, such as chronic obstructive pulmonary disease (COPD), pulmonary emphysema, or interstitial lung diseases (e.g., pulmonary fibrosis), would similarly lead to alterations in modulation power, as these conditions were shown to cause structural and functional changes in lung tissue. This highlights the necessity for a further evaluation of modulation power in human lung tissue, with particular attention to individual or patient-specific factors. As outlined in Section 9, novel and diversified approaches are required to address this lack of data.

## 6. Implementation of Lung Modulation Methods in Treatment-Planning Systems

There are currently two approaches that enable the reproduction of the Bragg peak degradation for dose calculations engines. Both methods employ a manipulation of input data in certain ways: The first approach, a statistical one, is based on manipulating the image or geometry data and was proposed by Baumann et al. [11]. The modulation is achieved by changing the physical density of the image voxels within a certain volume of interest (VOI), for example the segmented lung region. The CT is duplicated numerous times (∼100 times). For every duplicate, each voxel within the VOI is replaced by a voxel of a different density. The assigned new density for each voxel follows a distribution in the form of a Gaussian distribution. The parameters μ and σ of the Gaussian are determined by the original density of the voxel and the modulation power one wants to apply. However, for small volumes (e.g., 2 mm voxel size) the normal distribution for small path lengths ($P\left(t^{\prime }\right)$) will have non-negligible contributions of negative densities (see Figure 5). To mitigate this problem, Ringbæk et al. [12] introduced an extended continuous Poisson distribution ($EP^{*}\left(t^{\prime }\right)$) to describe the modulation for small volumes. The main characteristics of $EP^{*}\left(t^{\prime }\right)$ are its transition from a discrete to a continuous function, the absence of negative contributions, and the high weight assigned to the water-equivalent thickness at $t^{\prime }=0$. However, the function convoluted n-times by itself results in the same distribution as a convolution of Gaussian distributions $P\left(\sigma ,N\cdot t\right)$ (see Equation (Equation 6)). A detailed description of the function can be found in [12] and is schematically displayed in Figure 5.
(6)$$\underset{N-times}{\underset{︸}{EP^{*}\left(t^{\prime }\right)*EP^{*}\left(t^{\prime }\right)*\ldots *EP^{*}\left(t^{\prime }\right)}}=EP^{*}\left(\sigma ,N\cdot t\right)\approx P\left(\sigma ,N\cdot t\right)$$

With this definition and a given modulation power $P_{\mathrm{mod}}$, any image data can now be manipulated to introduce a Bragg peak degradation. Baumann et al. [11] used this implementation for Monte Carlo dose calculations, whereas Ringbæk et al. [12] used this method with a deterministic pencil beam algorithm.

The second implementation employs a manipulation of the Bragg peak curves to mimic the lung modulation effect. Winter et al. [13] and Paz et al. [14] implemented an on-the-fly convolution of the unperturbed Bragg curve $b_{0}$ (e.g., base data of the TPS) with a Gaussian convolution kernel $P\left(t^{\prime }|\sigma ,t\right)$ (Equation (Equation 2)). On-the-fly indicates that the convolution kernel depends on the total path length through the modulating material and $P_{\mathrm{mod}}$. For each voxel upstream any lung voxel, the total path length has to be calculated using a ray-tracing algorithm. Subsequently, for each dose point, a unique convolution has to be explicitly calculated. This makes it quite time-consuming depending on the depth in the patient, the dose grid resolution, and target volume. Figure 6 shows the result of a running convolution in a simplified phantom case. The phantom consists of 20 mm of water followed by lung tissue. In the first 20 mm of water, the curves of the unmodulated reference and the modulated beam are the same. As the beam enters the lung tissue and the modulation is taken into account, the curves begin to diverge.

Winter et al. [13] used matRad, an open-source treatment planning toolkit [48], as a treatment planning software. They furthermore used a special Gaussian parameterization of the depth–dose curves as described by Bangert et al. [49,50]. The implementation from Paz et al. [14] was established in GSI’s in-house treatment planning software TRiP98 [51]. It was the first implementation to also take into account the effects of lung modulation on the biological dose. The same concept as the modulation of the depth–dose curve was adopted for the dose-averaged α, β, and LET distributions. In comparison to a voxelwise modulation of the full particle energy spectra, this was a major computation time improvement and showed comparable results (see Figure 7).

This allows for biological effects to be taken into account, thereby enabling biological dose calculations. This is of major relevance for carbon ions but is getting more and more relevant as more ions, such as helium, find their way into the clinic [52].

Both implementations have their benefits. The image-based statistical approach enables Monte Carlo computations, whereas the second implementation enables a compensation of the modulation effect, as the degradation model can already be used in the optimization step and therefore directly compensates for the degraded dose, thereby regaining target coverage [14].

## 7. Investigation of Dose Uncertainties Due to Lung Modulation Effects

The method of reproducing the Bragg peak degradation by modulating the density of clinical voxels was adapted in a study by Flatten et al. [15] to investigate characteristics of lung modulation effects via a CT-based phantom study (see Figure 8).

These phantoms were designed to represent the anatomical situation for lung irradiation: The phantoms consisted of a 2 cm thick slab of water representing the chest wall, followed by 25 cm of lung tissue. Artificial tumors of different volumes and shapes were located at different positions within the lung. Tumor volumes were between 1 cm^3^ and 43 cm^3^, the depth in the lung was between 2 cm and 20 cm. The density of the lung tissue was set homogeneously to 0.26 g/cm^3^, which is the density of an inflated lung [53]. This scenario represents the clinical situation where the microscopic structure of the lung is not resolved and a more homogeneous density distribution is predicted in the CT image.

For each phantom, a proton treatment plan was optimized using the commercially available TPS, Eclipse V13.7 (Varian Medical Systems). For each plan, a single beam perpendicular to the entrance wall of the CT-based phantom was optimized. The lateral spot spacing was 60% of the FWHM of the beam spot and the energy spacing was between 1 MeV for small tumor volumes and 3 MeV for larger tumor volumes. All treatment plans were subsequently recalculated with Monte Carlo simulations in two different scenarios: In the first scenario, the density of the lung tissue was kept at 0.26 g/cm^3^. This non-modulated case corresponded to the prediction from the TPS that did not consider the heterogeneity of the lung tissue. In the second simulation, the density of each voxel within the lung was modulated according to modulation power values of 250 μm, 450 μm, and 800 μm. The dose distribution from this modulated case corresponded to the dose distribution as it would occur in the patient due to the lung modulation. By comparing the dose distributions, the effects of the lung modulation could be extracted and analyzed. The choice of modulation power values was again driven by the results from Witt [8] as no further data were available at that time.

The effects of lung modulation were quantified by the decrease in the average dose $D_{\mathrm{mean}}$ deposited in the PTV (Figure 9). It was shown that the Bragg peak degradation led to an underdosage of the PTV, especially at the distal end. The effects increased with an increasing depth of the tumor in the lung since the distance particles were traversing through heterogeneous lung tissue increased. In addition, the underdosage of the PTV increased with decreasing tumor volume. For a modulation power of 450 μm, which is the average modulation power as measured by Witt [8], the minimal underdosage was smaller than 1% in terms of the reduction in $D_{\mathrm{mean}}$ for large tumors located near the chest wall and up to 8% for small tumors located at large depths. For a modulation power of 800 μm, the effects were larger with an underdosage of up to 15%. Concerning the shape of the tumor, it was shown that it was not only the volume that defined the dimensions of the underdosage but mainly the extension of the tumor in beam direction: the smaller the tumor in the beam direction, the larger the underdosage of the tumor.

In a second Monte Carlo based study [16], clinical proton treatment plans of NSCLC patients were investigated to quantify the effects of the Bragg peak degradation for clinical cases and to give a conservative approximation of the lung modulation effects. For five patients with tumor volumes between 2.7 cm^3^ and 46.4 cm^3^ (Note that small tumor volumes might not benefit from particle therapy compared to photon-based SBRT [1]; however, these small tumor volumes were investigated anyway since such small tumors have been treated at different centers [54,55,56] justifying an investigation), proton treatment plans were optimized. The patients were selected to cover a large variety of tumor volumes and to have tumors located at the center of the lung as well as tumors located near soft tissue or organs at risk. No tumors were located near the chest wall to always have lung tissue in the beam. The patients were originally treated with photons and retrospectively re-planned with protons for the study.

For each patient, three proton treatment plans were optimized, while each treatment plan consisted of a single field coming from either 0°, 270°, or 315°. The plans were chosen to be simple in order to highlight the effects from the Bragg peak degradation. The use of different beam directions enabled scenarios with different depths of the tumor in the lung. In particular, those plans are relevant when large volumes of lung tissue are irradiated in order to give an upwards estimation of lung modulation effects. The depths of the tumors in lung were between 1.5 cm and 12.2 cm. All treatment plans were optimized in Eclipse v.13.7 (Varian Medical Systems). The total prescribed dose was 30 Gy (RBE), and the only planning objective was to deliver at least 95% of that prescribed dose to at least 98% of the PTV. The PTV was the CTV plus an isotropic margin of 3 mm. The distal spot spacing was 3 mm, and the lateral spot spacing was 0.45 times the FWHM of the beam spot. The treatment plans were optimized on static CT data ignoring movements of the anatomy due to respiration.

Analogue to the study from Flatten et al. [15], all treatment plans were subsequently recalculated in Monte Carlo simulations with and without modulating the density of the lung voxels. The effects of the lung modulation were investigated for modulation power values of 100 μm, 250 μm, 450 μm, and 800 μm. Again, it was shown that the lung modulation led to an underdosage of the PTV if not accounted for during treatment planning. The effect increased with an increasing modulation power, an increasing depth of the tumor in the lung, and a decreasing tumor volume. For a modulation power of 800 μm, the underdosage in terms of the average dose $D_{\mathrm{mean}}$ in the PTV was at most 5%. For a more realistic modulation power of 450 μm, the underdosage was at most 3% and only 1% on average. Additionally, it was shown that for all investigated treatment plans, the passing rate of the Gamma index analysis (3%/1 mm) was a minimum of 90.4% and 96.8% on average for a modulation power of 800 μm. For a modulation power of 450 μm, the passing rate was a minimum of 93.1% and 98.5% on average. The authors further investigated the shift of isodose lines [57,58] for 95%, 80%, and 20% of the prescribed dose (see Figure 10). For the modulated case representing the dose distribution in the patient, the regions enclosed by the 80% and 95% isodose lines were smaller compared to the non-modulated case, demonstrating the underdosage of the target volume. The range uncertainties of the isodose lines in lung tissue were at most 8 mm for the 95% isodose lines and 10 mm for the 85% isodose lines.

The range uncertainties in soft tissue were smaller with only 4 mm and 3 mm, respectively. The 20% isodose lines reached farther for the modulated case demonstrating a potential overdosage of distal normal tissue. The range uncertainties were within 5 mm in lung tissue and 2 mm in soft tissue. No significant increase in the dose deposited to organs at risk could be seen in any of the investigated treatment plans.

Besides the studies investigating the lung modulation effects with the help of Monte Carlo simulations, Winter et al. [13], Ringbæk et al. [12], and Paz et al. [14] investigated these effects with the help of deterministic dose calculation algorithms. Winter et al. [13] retrospectively investigated the effects of lung modulation in proton beams for ten patients who were originally treated with photons. The volume of the PTV ranged from 10 cm^3^ to 1480 cm^3^. For each patient, one intensity-modulated proton treatment plan was optimized. The number of fields used for each plan varied between a single field up to five fields. Treatment plans were optimized with syngoRT planning (Siemens Healthineers, Erlangen, Germany). Subsequently, these plans were recalculated with matRad, an open-source treatment planning toolkit [48]. Two different modulation power values were investigated: 256 μm as a realistic modulation power and 750 μm for an estimation of maximum effects.

The patients were divided into three groups:Group L: target volumes in the middle of the left or right lung;Group C: target volumes located near the mediastinum (centrally);Group W: widespread target volumes that cover a large area within the lung.

Dose uncertainties due to lung modulation effects were investigated in terms of $D_{95}$ (minimum dose that is received by 95% of the CTV or PTV), median dose $D_{50}$, and $V_{\mathrm{under}}$, the volume of the PTV receiving less the 95% of the prescribed dose.

For a modulation power of 256 μm and patients of groups C and W, relative changes in the $D_{95}$ for CTV and PTV as well as $D_{50}$ due to lung modulation effects were smaller than 0.8%. For patients in group L, the changes were up to 1.6%. For all cases, $D_{50}$ decreased, demonstrating the underdose of the target volume due to the effects of lung modulation.

The change in $V_{\mathrm{under}}$ was small for patients of group C. In one case, the lung modulation even led to a decrease in $V_{\mathrm{under}}$, as cold spots from the original treatment plan were smoothed out due to the lung modulation. For patients with small tumors, the change in $V_{\mathrm{under}}$ was 20–26%.

The effects for a modulation power of 750 μm were larger and were shown for one exemplary patient. The relative change in $V_{\mathrm{under}}$ increased from 25% for a modulation power of 256 μm to 44% for a modulation power of 800 μm. The underdosage in terms of a relative change in $D_{50}$ increased from −1.5% to −4.4%.

Paz et al. [14] investigated retrospectively one sample carbon ion patient case which was originally treated with photons. Again, two modulation power values (256 μm and 450 μm) were investigated. The CTV volume was small with 1.6 cm^3^ and was situated in the left superior lung. The treatment planning was performed with carbon ions, and it was the first study to report the effect of lung modulation on a biologically calculated dose and the first study to also compensate for the lung modulation by implementing it in the optimization stage. The impact of lung modulation was assessed based on $D_{95}$ and $V_{95}$ values of the CTV as well as the mean dose to normal tissue in a 13 mm thick spherical shell around the CTV.

The $D_{95}$ and $V_{95}$ of the modulated case was reduced by 1.5% and 1.6% for $P_{\mathrm{mod}}$ = 256 μm and 3.2% and 3.7% for $P_{\mathrm{mod}}$ = 450 μm, respectively, compared to the unmodulated reference (see Figure 11). The optimized treatment plan was able to achieve the same results in terms of target coverage as the unperturbed reference. However, due to the increased distal fall-off, this could only be achieved by simultaneously increasing the dose to the surrounding healthy tissue. The increase in the mean dose to the 13 mm shell around the CTV was 8.1% ($P_{\mathrm{mod}}$ = 256 μm) and 10.5% ($P_{\mathrm{mod}}$ = 450 μm), respectively.

Ringbæk et al. [12] also investigated lung modulation effects with the help of analytical dose calculation algorithms; however, modulation effects were not implemented by manipulating base data, but by modulating the density of lung voxels within the CT data. In addition to the investigation of the effects of lung modulation for proton beams, Ringbæk et al. [12] also investigated the effects for carbon ion treatment plans although only for the absorbed dose. In a first part, a planning study on spherical tumor geometries of different sizes located at different depths in lung tissue was conducted for carbon ion plans. The modulation power was set to 400 μm, and for each geometry, a single field plan with 2 mm lateral spot spacing and 3 mm distal spot spacing was optimized. The results from Flatten et al. [15] could be reduced (increasing effect for increasing depth in lung and decreasing tumor volume).

In a second part, treatment plans for two exemplary patients were investigated. One patient had a deep-seated tumor, the other a tumor located centrally in the lung. Both tumor volumes were relatively small. For both patients, multiple proton and carbon ion plans were optimized, each plan consisting of one or two fields. The planning objective was to deliver at least 95% of that prescribed dose to at least 98% of the PTV. Modulation power values between 300 μm and 600 μm were investigated. It was shown that the lung modulation effects were larger for carbon ion plans compared to proton plans, since carbon ions generally develop a sharper Bragg peak and hence are more sensitive to changes in the Bragg peak width. For carbon ion plans, the homogeneity index (HI) increased by up to a factor of three. The decrease in the $D_{98}$ was as much as 14%.

## 8. Clinical Context of Dose Uncertainties Due to Lung Modulation Effects

In general, all studies show that lung modulation effects lead to an underdosage of the target volume. The effects strongly depend on the modulation power and patient geometry and increase with modulation power, the length of penetrated lung tissue, and a decreasing extension of the tumor in the beam direction. The effects for proton plans are in the order of some percent while range uncertainties are at most several millimeters in lung tissue and even smaller in soft tissue. Typically, particle therapy is connected to general range and dose uncertainties. Range uncertainties in proton therapy arise from uncertainties in commissioning measurements (±3 mm), patient setup (±7 mm) and uncertainties in the dose calculation in treatment-planning systems (±2 mm) [59]. Additionally, range uncertainties are due to the conversion from X-ray HU units from CT images to stopping power values for protons (±1% of the range) and uncertainties in the modeling of the biological effectiveness (∼0.8% of the range or ∼3 mm) [59,60]. Next to range uncertainties, dose uncertainties arise from anatomical changes in the patients and especially for the treatment of lung cancer patients, from motion. Anatomical changes can lead to an undercoverage of the target volume from 2 GyE to 12 GyE for proton treatment of locally advanced NSCLC, which corresponds to relative deviations of 3% to 18% in terms of $D_{99}$ [61]. Interplay effects due to motion can lead to a reduction in the target coverage in the order of 10% [62]. Compared to general range and dose uncertainties connected to proton therapy, effects of the lung modulation are comparable in the case of range uncertainties, and significantly smaller in the case of dose uncertainties for the treatment of NSCLC. Hence, lung modulation effects can be tolerated to a certain extent in the clinical context of range and dose uncertainties in proton therapy. However, one has to keep in mind that lung modulation effects are systematic and always lead to an underdosage of the target volume, while range uncertainties are of statistical nature and can potentially be compensated over the course of several fractions. The same accounts for interplay effects due to target motion which typically decrease with an increasing number of applied fractions [30].

The effects for carbon ions are more pronounced; however, an investigation of larger patient cohorts is needed to better quantify these effects. In general, lung modulation effects can be reduced by adapting the treatment plan in order to choose beam directions where the penetrated length of lung tissue is minimal.

## 9. Determination of Patient-Specific Modulation Power

The dosimetric investigation of lung modulation effects has shown that the underdosage of the target volume strongly depends on the modulation power of lung tissue. As described in Section 4, data for the modulation power are only available for ex vivo lung tissue, while the preparation of lung tissue samples can lead to an adulteration of the determined modulation power due to a loss of water or blood as well as mechanical destruction of the lung structure. Hence, to minimize dose uncertainties, the patient-specific modulation power is needed, at best spatially resolved, since the modulation power depends on the size of the structures and hence on the position within the lung.

A first solution is using proton radiography [63]. By penetrating the lung with protons and measuring depth–dose distributions behind the patient, the modulation power can be determined in principle. However, as long as only one projection is used, the integrated modulation power is determined as in the case of the measurement with porcine lung tissue samples in the studies by Witt and Burg et al. [7,8]. In order to obtain 3D information, a proton tomography would be needed. Currently, this is not a feasible solution for the clinical routine, especially due to the high dose deposition. However, for the sole intention to determine a 3D distribution of modulation power values, one does not need to measure depth–dose distributions. In principle, it would be sufficient to measure an energy distribution distal to the penetrated lung tissue, as the modulation power can also be derived from that information. To measure an energy distribution, less particles are needed compared to a dose measurement. Still, the clinical implementation of proton radiography or even proton tomography will take time.

Other promising tools that can be used to obtain information about the structure of the lung tissue is the implementation of dark-field phase-contrast CT imaging [64,65,66]. Dark-field phase-contrast CT imaging is able to provide a high soft tissue contrast as well as a high spatial resolution. Recent studies [67,68] have shown that patients with chronic obstructive pulmonary disease (COPD) can be staged, and changes in lung tissue due to COVID-19 can be detected using dark-field phase-contrast imaging, indicating that this technique might be used to establish a connection from image data to the modulation power of lung tissue.

A different approach was proposed by Hranek et al. [46] and Flatten et al. [10]: In general, the modulation power of lung tissue depends on the structure size. These structures are not sufficiently resolved in CT images, but different structures are merged into one CT voxel. Depending on the size of the structures, the number of structures merged into one voxel varies, which should have an influence on the distribution of HU values. Hranek et al. [46] showed that the histogram of HU values was different for different heterogeneous materials with modulation power values between 100 μm and 1000 μm. However, no clear correlation between HU histogram and modulation power could be established. Flatten et al. [10] used a mathematical model to derive a connection between structure size and CT histogram. It was shown that when fitting the CT histogram with a normal distribution, the modulation power could be derived from the mean value μ and the width σ of this normal distribution. Using inorganic heterogeneous materials, this model was tested extensively: modulation power values of the material samples were determined with in-beam measurements. Additionally, CT images of these material samples were acquired, and histograms of CT values created. A clear correlation between CT histograms and modulation power could be established. Subsequently, the model was verified using porcine lung samples. Modulation power values were derived using the histogram analysis and compared to values of in-beam measurements. The agreement was within 150 μm for all porcine lung tissue samples.

The big advantage of this model by Flatten et al. [10] is that by using already available clinical CT images, the modulation power can be estimated for each patient specifically and for each region of the lung separately. In particular, this method does not need any further dose deposition in the patient since CT images are already available, and no further imaging technique is needed. In principle, the only step that has to be taken before this method can be used clinically is a calibration of the CT scanner since the connection between CT histogram and modulation power severely depends on the characteristics of the CT scanner. The uncertainty of the modulation power determined via the CT histogram analysis is roughly 150 μm, which is sufficiently accurate since dosimetric studies showed that changes in the modulation power of ∼100 μm resulted in changes in the dose uncertainties of about 1%.

## 10. Conclusions

In conclusion, the fine heterogeneous structure of lung tissue leads to a degradation of the Bragg peak and hence a dose modulation in particle treatment plans for lung cancer patients. If not accounted for during treatment-planning optimization, these effects lead to an underdosage of the target volume and overdosage of distal normal tissue. Due to the restricted resolution of clinical CT scanners, these lung structures are not fully resolved, making a direct consideration during treatment-planning almost impossible.

In recent years, this problem was solved by quantifying the lung modulation effects by implementing the material’s characteristic “modulation power” ($P_{\mathrm{mod}}$). This quantity can be calculated if the structure of a heterogeneous material is known exactly or can be determined with in-beam experiments for materials with an unknown structure, such as lung tissue. Several measurements were performed for ex vivo lung tissue samples; however, due to the preparation of these samples, measurements results were connected to a large uncertainty.

Using the modulation power, a Monte Carlo based method was developed to consider lung modulation effects on clinical CT images by applying a density modulation to the voxels associated with the lung. Using this method, dose uncertainties due to lung modulation effects were investigated for CT-based phantoms as well as clinical treatment plans for proton and carbon ion irradiation. While dose uncertainties for proton beams were on the percent level, and hence tolerable in the current clinical context, dose uncertainties for carbon ion beams were larger. It was shown that the underdosage of the target volume increased with modulation power, penetrated lung tissue and decreasing the extension of the tumor in the beam direction.

A first proof of principle was presented on how to include lung modulation effects into the treatment-planning process in order to minimize dose uncertainties.

In particular, the determination of patient-specific modulation power values will be a key element in the future. As proposed, the modulation power can be estimated via a HU histogram analysis. This approach certainly is a feasible tool to include the consideration of lung modulation effects in the clinical treatment-planning workflow.

Another issue that will need further investigation for carbon ion beams is the modulation of biological dose. Paz et al. [14] showed that not only the physical dose was modulated due to the heterogeneous lung tissue, but a modulation effect could also be observed for the biological dose. In particular, a widening of the RBE distribution was observed beyond the distal edge, potentially being harmful for distal normal tissue.

## Figures and Tables

**Figure 1 cancers-16-03598-f001:**
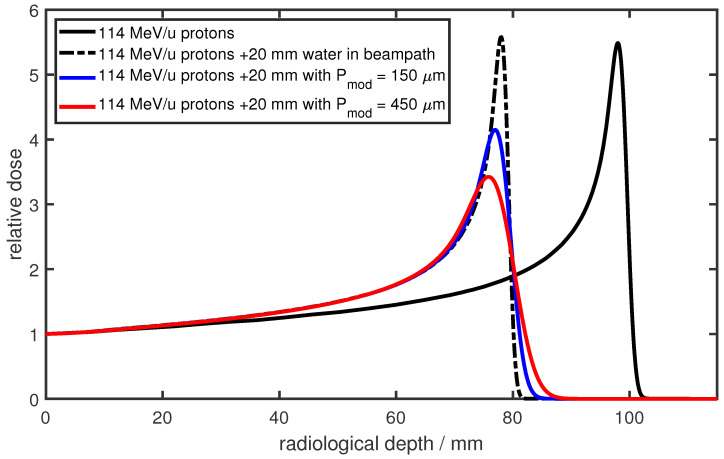
Depth dose distributions for 114 MeV protons in water. The solid line is the reference Bragg peak. The dashed line shows the same proton beam with an additional 20 mm of water in the beam path. In red and blue, again, a 114 MeV/u proton beam but with an additional 20 mm of modulating material with a modulation power of 150 μm (blue) and 450 μm (red) in the beam path. The modulating material (e.g., lung tissue) leads to a shift in the Bragg peak due to the additional material in the beam line and to a broadening of the Bragg peak due to the heterogeneous structure.

**Figure 2 cancers-16-03598-f002:**
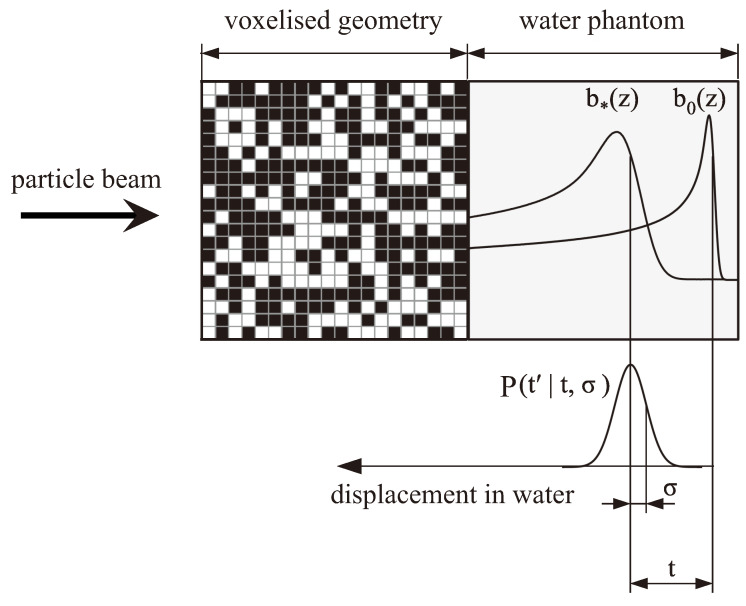
Schematic description of the convolution. An unperturbed reference curve $b_{0}\left(z\right)$ is scored in a water phantom with no additional material in the beam line. Subsequently, the modulated dose distribution $b_{*}\left(z\right)$ in the water phantom when the heterogeneous, voxelized geometry is in the beam line can be derived by convolving $b_{0}\left(z\right)$ with $P\left(t^{\prime }|\sigma ,t\right)$. $P\left(t^{\prime }|\sigma ,t\right)$ gives the probability that a specific path through the voxelized geometry has a water-equivalent thickness of $t^{\prime }$. *t* gives the average water-equivalent path length and σ is a measure of the strength of the degradation [11].

**Figure 3 cancers-16-03598-f003:**
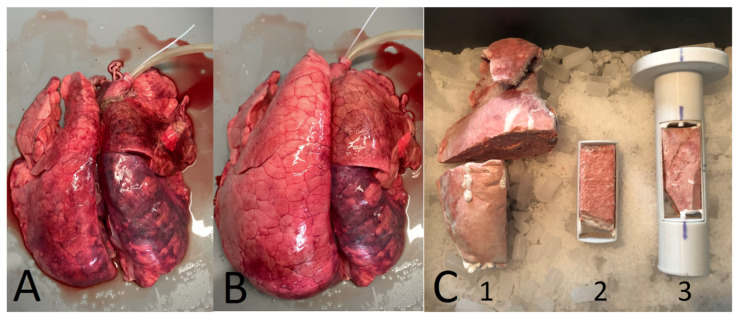
Part (**A**) shows a lung before and picture (**B**) after ventilation. On the right side in (**B**), it can be seen that the lower part of the left lung could not be ventilated correctly. Part (**C**) shows the cutting process from the frozen lung to the final lung sample for the measurements. C1: cutting the lung to sample size. C2: fixation in the sample frame. C3: final lung sample within the sample holder for measurements. From Burg et al. [7].

**Figure 4 cancers-16-03598-f004:**
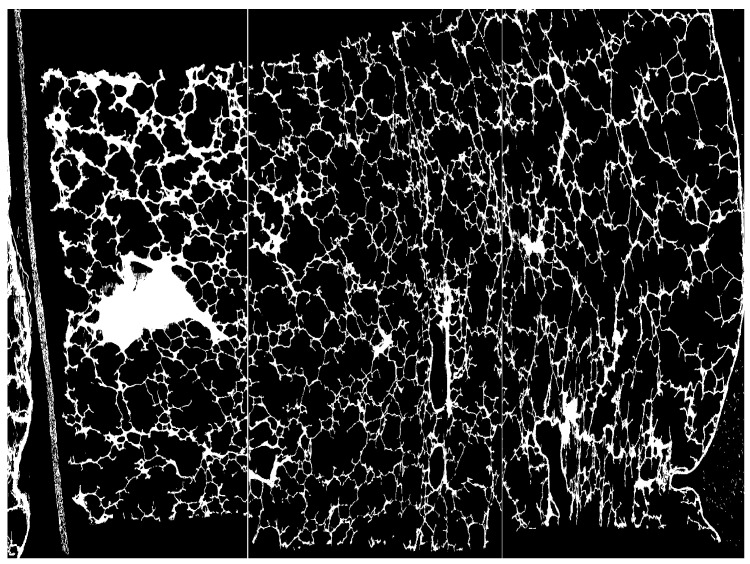
Micro-CT of human lung sample (punch of 8 mm diameter and 10 mm height) with a spatial resolution of 4 μm. The samples show a wide variation in alveoli sizes as well as hyperdense and hypodense areas originating from larger blood vessels or larger bronchioles. A detailed description of sample preparation can be found in [47]. The image postprocessing is described in [8]. Image from Witt [8].

**Figure 5 cancers-16-03598-f005:**
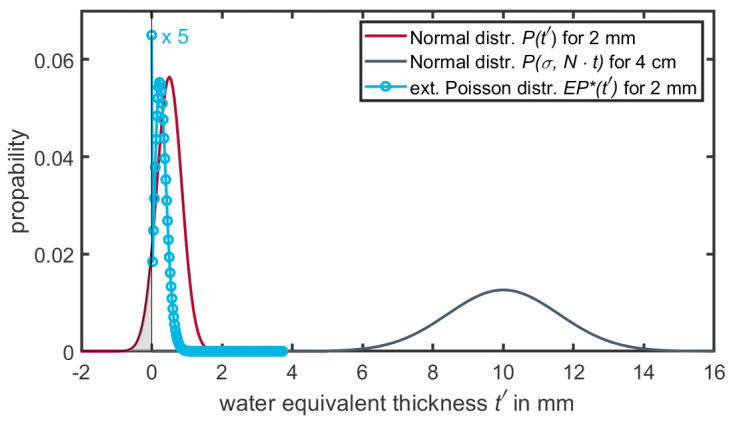
Extended Poisson distribution ($EP^{*}\left(t^{\prime }\right)$) and normal distributions ($P\left(t^{\prime }\right)$) for small volumes (e.g., 2 mm voxel). The normal distribution has non-negligible negative contributions (gray area). The Poisson distribution for a 4 cm path length ($EP^{*}\left(\sigma ,N\cdot t\right)$) is derived by a 20-time convolution of the Poisson distribution ($EP^{*}\left(t^{\prime }\right)$) and is indistinguishable from the normal distribution for 4 cm ($P\left(\sigma ,N\cdot t\right)$). The weight at x = 0 for $EP^{*}\left(t^{\prime }\right)$ is scaled by a factor of 0.2 for better visualization.

**Figure 6 cancers-16-03598-f006:**
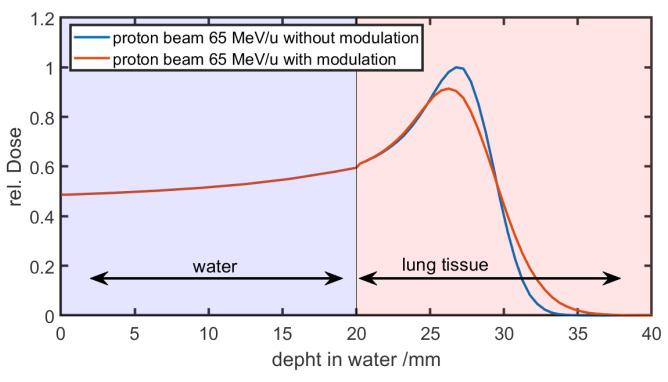
Modulated and unmodulated monoenergetic Bragg peaks of 65 MeV/u protons in a simplified phantom case. The phantom consists of 20 mm of water followed by lung tissue. In the first 20 mm where no modulation takes place, the curves are identical. When the beam enters the lung tissue, the curves begin to diverge, as the modulation increases with penetration depth in lung tissue.

**Figure 7 cancers-16-03598-f007:**
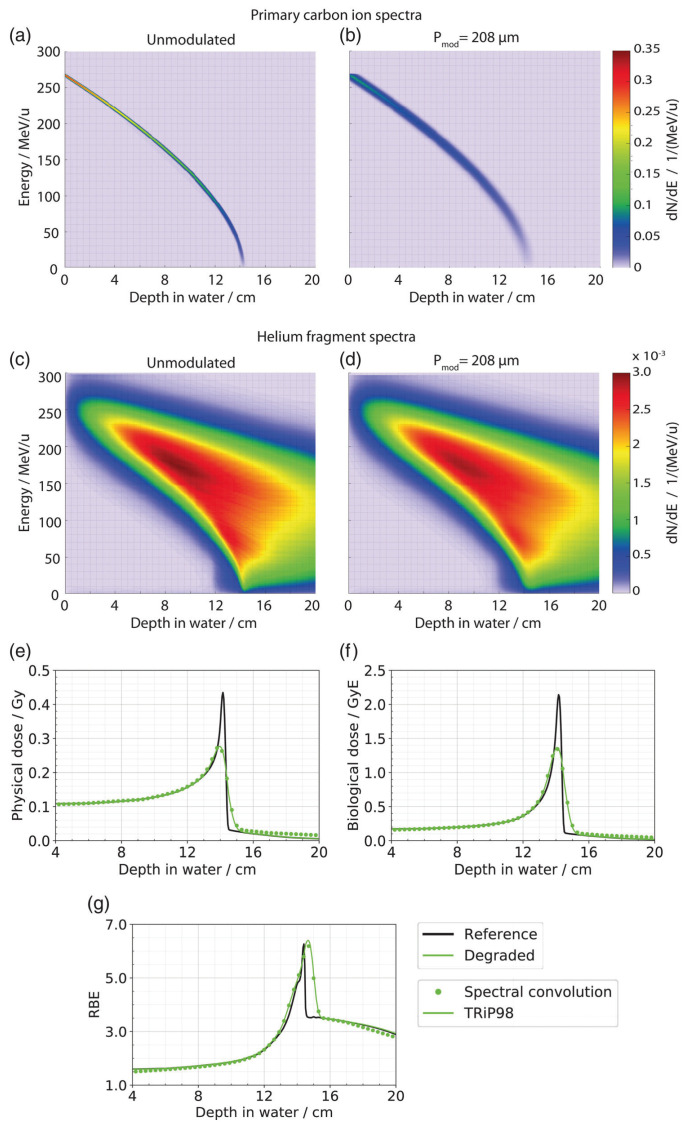
(**a**,**b**) Spectral distribution from primary carbon ions and (**c**,**d**) helium fragments with and without modulation for an incident 270.55 MeV/u carbon ion beam. (**e**) The physical dose, (**f**) biological dose, and (**g**) RBE distributions resulting from the spectral modulation were computed in Matlab (green circles). For comparison, the corresponding predictions from the FFT based TRiP98 implementation are superimposed as solid lines. From Paz et al. [14].

**Figure 8 cancers-16-03598-f008:**
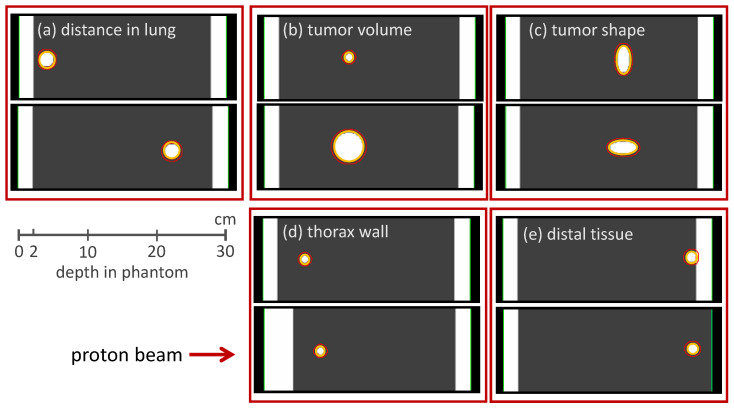
Phantom design with its major variations used in the study from Flatten et al. From left to right, different depths of the tumor in the lung and different tumor volumes and shapes are visible, as well as a change in the thickness of the proximal and distal water slabs. The yellow lines enclosing the tumor display the solid GTV, and the larger red line surrounds the PTV, respectively. From Flatten et al. [15].

**Figure 9 cancers-16-03598-f009:**
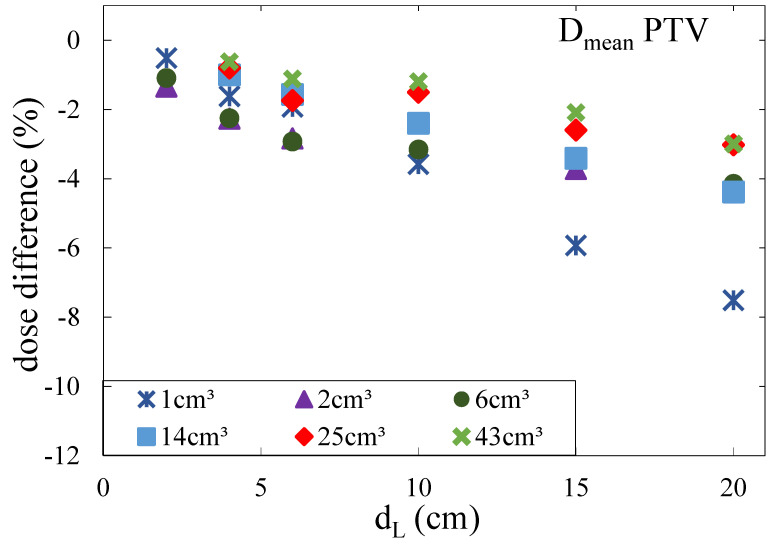
Relative effect of the Bragg peak degradation of the dose distribution depending on the tumor depth in lung tissue with a modulation power of 450 μm for the mean dose ($D_{\mathrm{mean}}$) of the PTV. Different markers indicate the tumor volume. From Flatten et al. [15].

**Figure 10 cancers-16-03598-f010:**
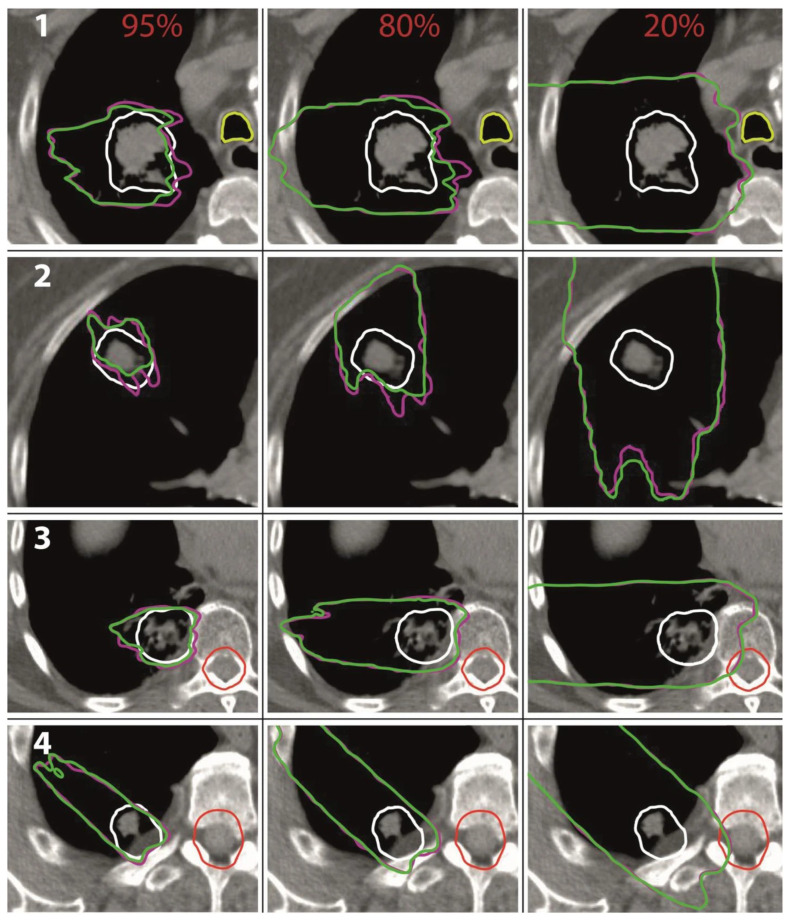
Isodose lines for 95, 80, and 20% of the prescribed dose for four different patients (1–4), in pink for the non-modulated case and in green for the modulated case based on a modulation power of 800 μm. In the first, middle, and right columns are the 95%, 80%, and 20% isodose lines. Different patient cases are marked in white numbers. The CTV is marked in white, the trachea in light green, and the spinal cord in red. From Baumann et al. [16].

**Figure 11 cancers-16-03598-f011:**
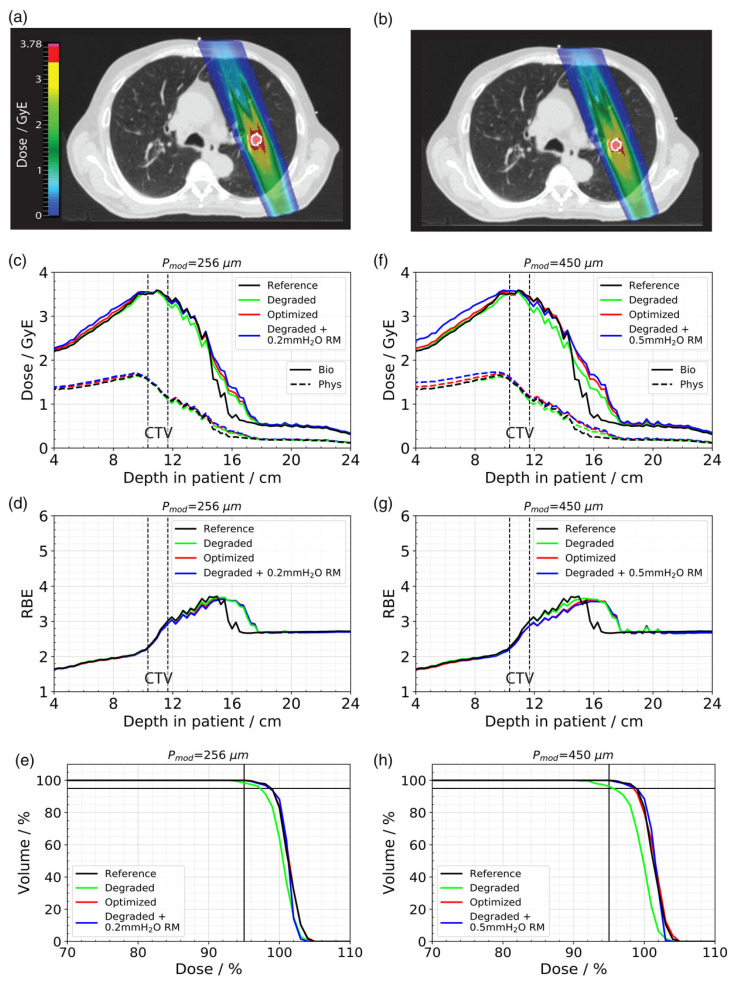
Comparison of the 4D-dose distribution from (**a**) unmodulated and (**b**) modulated ($P_{\mathrm{mod}}$ = 450 μm) treatment plans. The CTV contour is outlined in white. Biological and physical depth–dose profiles, RBE distributions, and DVHs from the reference, degraded, and optimized plans for (**c**–**e**) $P_{\mathrm{mod}}$ = 256 μm and (**f**–**h**) $P_{\mathrm{mod}}$ = 450 μm. D95 and V95 are indicated by solid black horizontal and vertical lines on the DVH plots, respectively. From Paz et al. [14].

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
