# Peer review of "On the Way to Accounting for Lung Modulation Effects in Particle Therapy of Lung Cancer Patients—A Review"

_cancers, 2024, doi:10.3390/cancers16213598_

Round 1

Reviewer 1 Report

Comments and Suggestions for Authors

Dear authors,

Thank you for submitting your comprehensive review on the impact of particle of particle therapy for treating NSCLC and the associated challenges posed by the modulation effects of lung tissue. This review is well-written and offers a detailed overview of the current state of research in this field, particularly regarding the degradation of the Bragg peak and its effects on dose distribution.

Several minor issues need to be addressed:

The authors included an image of Witt M. of his master thesis.  Are the rights to publish the same image in this publication obtained (permission)?

Please add all the abbreviations used in this manuscript e.g. COPD, DVH, IMRT etc

Overall, this review is well-written and presents a comphrensive summary of the key challenges and opportunities for particle therapy in the treatment of NSCLC. Only minor revisions are required before final acceptance of the manuscript.

Comments on the Quality of English Language

In total, the manuscript is well-written. However, some small changes should be performed e.g. abbrevation list.

Reviewer 2 Report

Comments and Suggestions for Authors

To attract this manuscript to the general public who reads cancer therapy and treatments in the general area, including boron and gadolinium neutron capture therapies, the authors must compare with such therapies to justify why their work is very significant and qualifies for publication in this journal. Therefore, my recommendation is to cite some more relevant publications, including book chapters, before further consideration. In addition, the authors' usage of English can be improved by getting it corrected by English experts in Europe.

Comments on the Quality of English Language

A minor editing of the English language can be done by the authors!

Reviewer 3 Report

Comments and Suggestions for Authors

The authors in the manuscript entitled "On the way to accounting for lung modulation effects in particle therapy of lung cancer patients - a review" well organized review article that describe the strategies to treat non-small cell lung cancer using particle therapy.

1. The abstract is well phrased

 2. The introduction needs to expand by explaining more about the background of the study 

3. For the readers, I suggest defining the spital dose and integral dose terms, where it comes first.

4. Please use colored figure (Figure 4) of micro-CT of human lung sample. 

5. All the sections well described for reader's understanding about the treatment planning and lung modulation effects of phototherapy.

In my view the review article is of reader's interest belong to oncology setup and researchers.
